# Remote Sensing Vision-Language Foundation Models without Annotations via Ground Remote Alignment

**Utkarsh Mall** [*,1,2], **Cheng Perng Phoo** [1,*], **Meilin Kelsey Liu**[1], **Carl Vondrick**[2],
**Bharath Hariharan**[1], **Kavita Bala**[1]
[1]Cornell University, Ithaca, NY     [2]Columbia University, New York, NY     [*]Equal Contribution
Correspondence: um2171@columbia.edu

## ABSTRACT

We introduce a method to train vision-language models for remote-sensing images without using any textual annotations. Our key insight is to use co-located internet imagery taken on the ground as an intermediary for connecting remote-sensing images and language. Specifically, we train an image encoder for remote sensing images to align with the image encoder of CLIP using a large amount of paired internet and satellite images. Our unsupervised approach enables the training of a first-of-its-kind large-scale vision language model (VLM) for remote sensing images at two different resolutions. We show that these VLMs enable zero-shot, open-vocabulary image classification, retrieval, segmentation and visual question answering for satellite images. On each of these tasks, our VLM trained without textual annotations outperforms existing VLMs trained with supervision, with gains of up to 20% for classification and 80% for segmentation. Our code, data, and other resources are available at: https://graft.cs.cornell.edu

## 1 INTRODUCTION

Our planet is constantly captured by an extensive array of remote sensors such as satellites or drones. These earth observation images enable the monitoring of various events on the earth such as deforestation, forest fires, and droughts so that rapid actions can be taken to protect our environment. While these images can shed light on various insights about our planet, the scale of such data is huge. This has prompted the development of automatic analysis models that could extract relevant information from a large amount of remotely sensed images. While useful, these models are often specialized and can only recognize a pre-defined set of concepts. Besides, they could be complex, decreasing their accessibility to experts outside of the domain of artificial intelligence.

Researchers developing automatic analysis methods for internet imagery encountered a similar problem a few years ago. One promising solution is to leverage large-scale vision-language models (VLMs) that are trained on millions or even billions of text-image pairs collected on the internet (Radford et al., 2021; Li et al., 2023). These models have demonstrated remarkable abilities to perform open-vocabulary recognition (Gu et al., 2022; Kuo et al., 2023) and enhance accessibility to non-AI experts (Alayrac et al., 2022; Surís et al., 2023).

It would be incredibly valuable for a range of applications to replicate the success of open-vocabulary recognition for satellite images as well, allowing an analyst to simply query, say, "Where are all the farmlands in the state of Massachusetts?" without requiring any new training or annotation for farms. However, building open-vocabulary vision-language models requires a large number of text-image pairs. This is difficult in remote sensing. Unlike internet images, which are often accompanied by captions or alt-text generated by their creators, satellite images are automatically generated by remote sensors with little to no human involvement and no corresponding text annotations. Prior work has attempted to annotate satellite images with textual annotations (Lu et al., 2017), but this process is expensive and requires expertise. As such, existing image-text datasets for remote sensing are almost four orders of magnitude smaller than the data used to train CLIP (10k

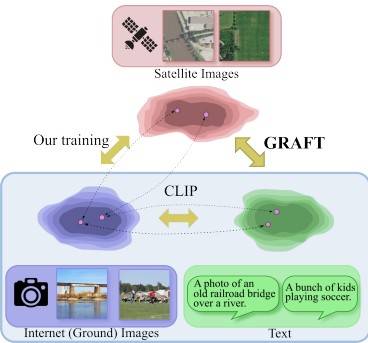

Figure 1: The key insight for building VLMs using GRAFT is to use internet ground images as an intermediary to connect satellite and text. By training a satellite image model to align its embedding with the CLIP embedding of co-located internet (ground) images, we transitively align text with satellite images, sidestepping the need for textual annotations for training remote-sensing VLMs.

vs. 400 million). This challenge motivates the question we answer in this paper: can we build a vision-language model for satellite images without textual annotations?

Our key insight is to leverage *internet images taken on the ground as an intermediary* between text and satellite images. A satellite image captures the condition of a particular location on Earth. The same location could be captured by humans on the ground with cameras. By leveraging the geotags associated with the ground images, we can easily link satellite images to them, creating a large dataset of ground-satellite image pairs. Coupled with pre-trained internet image encoders from CLIP, we use this data to train a vision transformer that can map satellite images to the CLIP encoder's feature space. We use a contrastive loss on these pairs. Since this feature space is shared by the CLIP text encoder as well, the satellite encoder allows an image-level textual understanding of satellite images, *completely sidestepping the need for textual annotations* (see Fig. 1).

Observing the fact that a satellite image can capture a much bigger physical space than a ground image (e.g., a ground image can only capture part of a lake but a satellite image can capture the whole lake), we further develop a text-to-patch retrieval model using our ground-satellite image pairs. Specifically, with the geotag associated with a ground image, we can identify the pixel location on the satellite image where the ground image is captured. We then construct a vision transformer that can output patch representations that align with the CLIP representation of the ground images. This model allows for not just classification but also localization: we show that we can use this representation to identify patches relevant to a particular textual query or perform text-to-image segmentation by leveraging foundational segmentation models such as SAM (Kirillov et al., 2023).

Since we leverage the alignment between ground images and remotely sensed images to construct vision-language models without textual annotations, we name our method **GRAFT** (**G**round **R**emote **A**lignment **f**or **T**raining). As shown in Fig. 2, GRAFT can perform classification, retrieval, semantic segmentation (in conjunction with SAM), and VQA (in conjunction with tools such as ViperGPT (Surís et al., 2023), all in a zero-shot manner. We extensively evaluate our VLMs on these tasks and demonstrate state-of-the-art zero-shot performance on various text-to-image retrieval (up to 20% relative improvement over baseline) and text-to-segmentation benchmarks (more than 80% relative improvement over baseline). Our contributions are summarized as follows:

- We introduce GRAFT which enables training remote sensing VLMs without any text annotations.
- To leverage GRAFT we collect two million-scale datasets of remotely-sensed images at different resolutions (1m for NAIP and 10m for Sentinel-2 images).
- Leveraging GRAFT with our dataset we develop *foundational vision-language model for satellite images* that can understand open-world concepts at two different resolutions and outperform prior arts on various image-level and pixel-level understanding tasks by a significant margin.
- We present a solution to the zero-shot VQA problem for satellite images by extending the ViperGPT framework with our VLMs.

## 2 RELATED WORKS

**Foundation Models for Remote Sensing Images.** Inspired by the recent success in internet imagery (Kirillov et al., 2023; Dosovitskiy et al., 2020; Liu et al., 2021; He et al., 2022; Kolesnikov

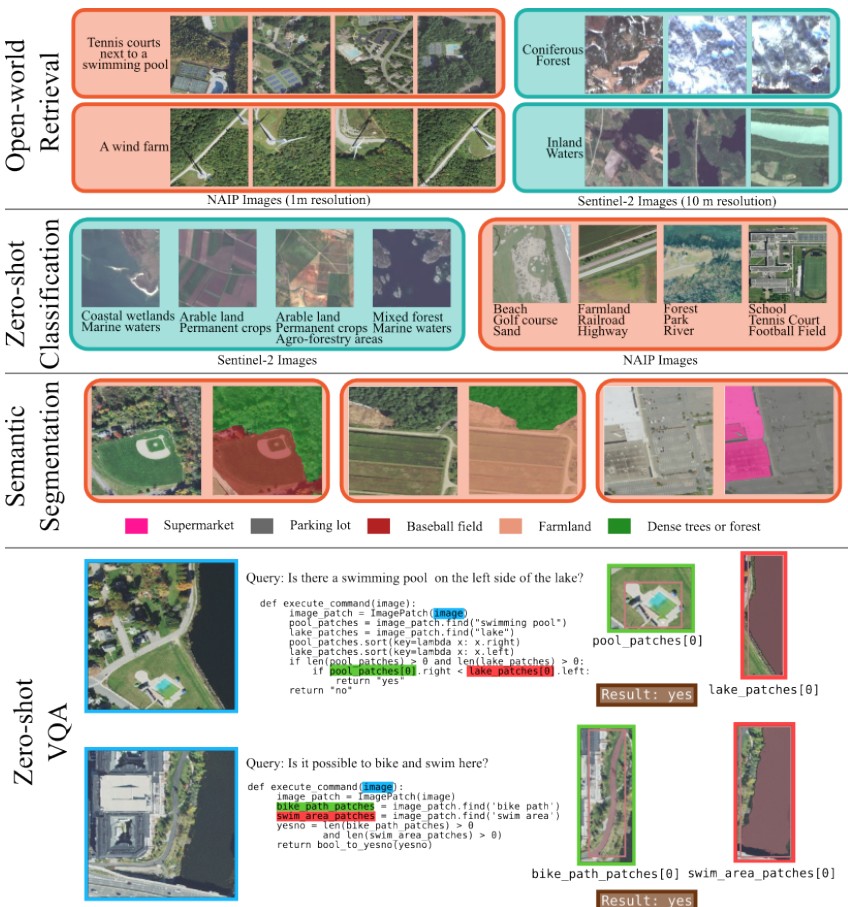

Figure 2: Zero-shot features of our model. GRAFT can perform image retrieval with open-world queries, and zero-shot classification for satellite images. Using other foundational models, we extend it to also perform semantic segmentation and zero-shot VQA. (Please view digitally to see details.)

et al., 2020; Dehghani et al., 2023), recent work has started exploring foundation models for remote sensing images that can then be fine-tuned for downstream tasks such as change detection. Some of these foundation models are supervised such as SatlasPretrain (Bastani et al., 2023) while others are based on self-supervised techniques such as contrastive learning (Manas et al., 2021; Mall et al., 2023) or masked image modeling (Jakubik et al., 2023; Sun et al., 2022; Cong et al., 2022; Fuller et al., 2022). These models are effective image encoders, but cannot perform open vocabulary recognition. Open vocabulary recognition can enable non-AI experts to analyze satellite data through natural interfaces such as retrieval or question answering, which is the focus of this work.

**Vision-and-Language Models in Remote Sensing.** Prior work has built VLMs for captioning or retrieval of satellite images (Yang & Newsam, 2010; Zhang et al., 2014; Lu et al., 2017) (see (Wen et al., 2023) for a comprehensive review). However, these models are trained on orders of magnitude less data (thousands of image-text pairs (Hu et al., 2023; Arutiunian et al., 2021) compared to billions for their internet counterparts), because text captions for satellite images cannot be crowd-sourced and must instead be curated. Recently, Zhang et al. also proposed a way to filter large image-text pair datasets to obtain satellite image-text pairs. However, many of the images come from a diverse range of sources and are not uniform. Fine-tuning from internet image models like CLIP(Radford et al., 2021) is another option(Liu et al., 2023; Al Rahhal et al., 2022), but limiting. In this work, we propose to sidestep the problem by leveraging ground images as an intermediary.

**Multi-modalility for better recognition.** We leverage ground images from the location where the remote sensing images are captured. Leveraging multiple complementary modalities has seen success in various domains. For instance, leveraging text and image has enabled open-vocabulary

object detection/segmentation in internet imagery (Gu et al., 2022; Ghiasi et al., 2022; Kuo et al., 2023). In remote sensing, using multi-spectral images or radar sensors allows a better understanding of the environment captured by a satellite (Cong et al., 2022; Zhao et al., 2023; Jakubik et al., 2023). The ground image modality has been also been explored by the community for geo-localization where the goal is to build models that could explicitly connect the remote sensing image and ground image modality (Lin et al., 2015; Toker et al., 2021; Berton et al., 2021; 2022b;a; Sun et al., 2023; Gu et al., 2023). Different from these works, we use ground images as an intermediary to connect remote sensing images with language — a novel uncharted problem domain to the best of our knowledge.

# 3 TRAINING VLMs WITHOUT TEXTUAL ANNOTATIONS

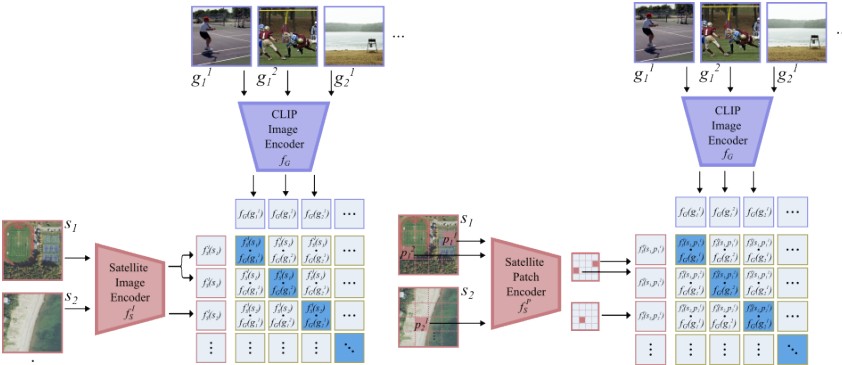

Figure 3: Training image-level VLM (left) and pixel-level VLM (right) with GRAFT. Note that for each satellite image there can be multiple ground images such as $\{g_1^1, g_1^2\}$ for $s_1$.

Our key insight is to create a vision-language model for satellite images by leveraging internet images as an intermediary between text and remotely sensed satellite images (see Fig. 1). In Sec. 3.1, we describe our training methodology GRAFT. In Sec. 3.2, we present our ground-satellite image pairs collection pipeline. Finally, we discuss how we can enhance our models with other foundation models to solve more general tasks for satellite images such as semantic segmentation and VQA.

## 3.1 GRAFT: GROUND REMOTE ALIGNMENT FOR TRAINING VLMs

We build two types of VLMs operating at different levels of understanding: image-level and pixel-level. Image-level models can be used to perform tasks that require understanding the satellite image as a whole such as text-to-image retrieval and zero-shot image classification; pixel-level models can be used when precise localization is required, such as for zero-shot segmentation and visual question answering (for questions like measuring the area of certain features). We note that many existing internet VLMs (Radford et al., 2021; Li et al., 2023) map internet text-image pairs to a common representation space. To build our desired VLMs we propose to build feature extractors that map satellite images to the same representation space. We assume access to a pre-trained internet VLM $(f_G, f_T)$ that maps an internet image-text pair to a common representation space (we use CLIP).

### 3.1.1 IMAGE-LEVEL VLMs

We wish to build an image-level feature extractor $f_S^I$ that maps a satellite image to the same representation space of $(f_G, f_T)$. We can use contrastive loss like CLIP to pull together corresponding pairs of satellite and ground images and push apart negative examples. However, the vanilla contrastive learning setup assumes that each data point in one modality (satellite image) maps to a single point in the other modality (internet images). Unfortunately, satellite images capture a larger area and have multiple ground images associated with them. Thus a new formulation is needed.

We posit that a satellite image's embedding should be close to *all* ground images captured in the area and be far from ground images for other satellite images. Specifically, for a batch of data

$$\mathcal{B} = \{s_i, \{g_i^j\}_{j=1}^{N_i}\}_{i=1}^{N_B} \tag{1}$$

where $s_i, i = 1, \ldots, N_B$ are satellite images and $g_i^j, j = 1, \ldots, N_i$ are the $N_i$ ground images taken in the geographical area captured by $s_i$. We capture our intuition using the following loss function:

$$\mathcal{L}^I(\mathcal{B}, f_S^I) = \frac{1}{N_B} \sum_{i=1}^{N_B} \frac{1}{N_i} \sum_{j=1}^{N_i} -\log \frac{\exp(f_S^I(s_i) \cdot f_G(g_i^j)/\tau)}{\sum_{a=1}^{N_B} \sum_{b=1}^{N_a} \exp(f_S^I(s_i) \cdot f_G(g_a^b)/\tau)} \qquad (2)$$

The outer sum is over satellite images in the batch, and the inner sum is over ground images for each satellite image. The numerator is the exponentiated similarity (we use cosine distance) between the satellite image $s_i$ and one of its ground images $g_i^j$, while the denominator contains the similarity of the satellite image $s_i$ to *every* ground image $g_a^b$ in the batch. $\tau$ is a temperature hyperparameter. Note that when $N_i = 1, \forall i$, Eq. (3) reverts back to the contrastive loss used in CLIP-like formulations. We use this loss to train $f_S^I$ while keeping $f_G$ frozen. See Fig. 3 for illustration.

Coincidentally, this loss function resembles the supervised contrastive learning (SCL) (Khosla et al., 2020), although the underlying problems are different. SCL is used for a single modality where examples of the same class are pulled closer to each other whereas we use the loss function to build an encoder for satellite images that aligns with a pre-trained representation of ground images.

### 3.1.2 PIXEL-LEVEL VLMS

Many satellite image understanding tasks such as segmentation require pixel-level localization. To enable pixel-level understanding, we turn to another source of information ignored in the previous section: the precise geographical location where the ground image $g_j^i$ is captured, which can be mapped to a pixel location $p_j^i$ in the corresponding satellite image $s_i$.

To leverage this signal, we assume a network architecture $f_S^P$ that can produce a feature vector $f_S^P(s)[p]$ for every pixel $p$ in the satellite image $s$. We implement $f_S^P$ using a ViT (Dosovitskiy et al., 2020) that produces feature vectors for non-overlapping patches of the satellite images. $f_S^P(s)[p]$ is then simply the output feature vector for the patch that contains pixel $p$. We train this feature extractor using a loss function similar to Eq. (3):

$$\mathcal{L}^P(\mathcal{B}, f_S^P) = \frac{1}{N_B} \sum_{i=1}^{N_B} \frac{1}{N_i} \sum_{j=1}^{N_i} -\log \frac{\exp(f_S^P(s_i)[p_i^j] \cdot f_G(g_i^j)/\tau)}{\sum_{a=1}^{N_B} \sum_{b=1}^{N_a} \exp(f_S^P(s_i)[p_i^j] \cdot f_G(g_a^b)/\tau)} \qquad (3)$$

to enforce the intuition that the pixel representation should stay close to its ground images in the representation space induced by the pre-trained VLM $(f_G, f_T)$. The above equation only provides sparse signals for training: the loss is computed only on pixel locations that have at least one ground image (see Fig. 3). Nevertheless, empirically we found that this sparse supervision suffices.

### 3.2 COLLECTING GROUND-SATELLITE IMAGE PAIRS

To perform our training, we need a dataset of ground-satellite image pairs. We collected two such datasets for two different kinds of remote sensing imagery: NAIP (U.S.G.S., 2022) (high resolution, with 1 meter per pixel) and Sentinel-2 (Drusch et al., 2012) (low resolution, with 10 meters per pixel). We describe our data collection process below.

**Ground Images:** We collect ground images from Flickr[1]. To obtain representative images from diverse regions (instead of just populous places), we uniformly select locations and sample non-duplicate images with accurate geo-tags (street-level accuracy). We remove indoor images using an indoor-outdoor classifier (a ResNet18 trained on SUN397 (Xiao et al., 2010)). While we found no discerning effect between using 'all' and 'outdoor' images, we filtered for ease of experimentation.

**Satellite Images:** We sample satellite images with their center at the geotags of the ground images. All the ground images whose geotags fall in this satellite image are assigned to it. Additionally, we do not sample satellite images using the already assigned ground images. As a result, we avoid high overlaps between satellite images (at least 112 pixels apart). We validate this sampling choice in Sec. 4.4. Fig. 4 shows the density of the image pairs in the US (left) and the world (right) with NAIP

---

[1]Flickr API: https://www.flickr.com/services/api

and Sentinel-2 respectively. To avoid over-representation, we randomly sample 25 ground images, if there are more than 25 of them. Our model uses 224×224 satellite images, however for better augmentation (rotation and translation without padding), we download 448×448 images.

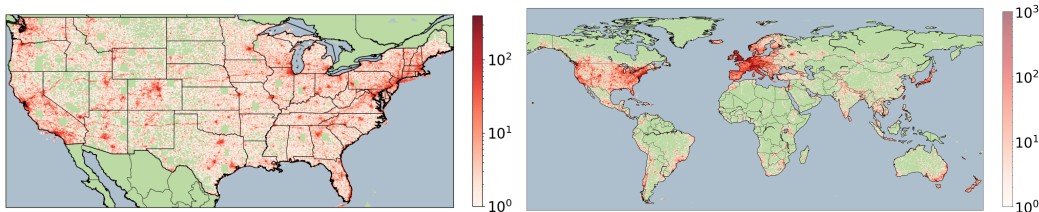

Figure 4: Frequency histogram of locations of samples in our internet image-NAIP image pair dataset (left) and in our internet image-Sentinel-2 image pair dataset (right).

In addition to location consistency between ground and satellite images, we also incorporate temporal consistency into Sentinel-2 data. Specifically, we collect the temporally closest images from the location of the internet images (containing $< 1\%$ of cloud). We cannot do so for NAIP since the revisit time of NAIP is much longer (once every 2 years *vs* once every 5 days for Sentinel-2). We make use of EarthEngine APIs[2] to obtain satellite imagery. Our dataset collection efforts yield 10.2 million pairs for NAIP and 8.7 million pairs for Sentinel-2 (also refer to Appendix A).

### 3.3 ENHANCING GRAFT VLMS WITH FOUNDATIONAL MODELS

While the performance GRAFT is impressive (see Sec. 4), we can further extend its capability using other foundation models and enable a wider range of applications:

**Zero-shot Image Segmentation.** While the pixel-level model can already be used to perform segmentation, we can improve its performance by leveraging bottom-up segmentation models like SAM Kirillov et al. (2023). To enhance the pixel-level model with SAM, we first select the highest-scoring patches using our model and then feed the center of the patch as point prompts to SAM.

**Visual Question Answering (VQA).** While GRAFT can be used to answer simple questions such as "Which satellite images contain a baseball field?", more nuanced questions may require sophisticated reasoning. To allow for more sophisticated questions, we couple our VLM with ViperGPT (Surís et al., 2023). ViperGPT uses an LLM to convert the natural language query into a program, which in turn makes calls to an open-vocabulary object detector. We replace the GLIP-based detector in Viper-GPT with our pixel-level GRAFT model. To produce a detection output, we threshold pixel-level scores and retrieve connected components to get instances, and then further refine each instance by using SAM as before. Further details are in the Appendix C.5.

### 3.4 IMPLEMENTATION DETAILS

We parameterize all our models using ViT-B/16 (Dosovitskiy et al., 2020). We also provide analysis on ViT-B/32 on a few tasks for fair comparison with baselines. All our models are initialized using the CLIP image encoder. We select hyperparameters using a validation set with NAIP resolution that we collected. This validation set is annotated using segmentation annotations from OpenStreetMaps from the same time as NAIP images as in (Bastani et al., 2023). The same set of hyperparameters is used for training the Sentinel-2 models. Please see Appendix C.2 for more information about this.

## 4 EXPERIMENTS AND RESULTS

In this section, we evaluate GRAFT on three kinds of tasks: image classification and retrieval, semantic segmentation and question answering. We ablate various design choices we have made.

---

[2]EarthEngine API: https://developers.google.com/earth-engine

Table 1: Zero-shot performance of Sentinel-2 image-level model on classification and retrieval benchmarks. Our model is significantly better on almost all the classification and retrieval metrics (red and blue indicate best and second best performance). We also compare our model against the 1-shot performance of remote sensing models that lack language capabilities.

| Model | Labeled Satellite Data | Input | Backbone | Classification | | Retrieval | | | |
|---|---|---|---|---|---|---|---|---|---|
| | | | | EuroSAT Acc. | BEN mAP | EuroSAT | | BEN | |
| | | | | | | mAP$^{100}$ | mAP$^{20}$ | mAP$^{100}$ | mAP$^{20}$ |
| CLIP | ✗ | Text | ViT-B/32 | 47.61 | 21.31 | 46.86 | 49.61 | 30.45 | 32.20 |
| CLIP | ✗ | | ViT-B/16 | *53.59* | 23.13 | 63.99 | 72.57 | 32.54 | 33.10 |
| CLIP-RSICD | ✓ | | ViT-B/32 | 45.93 | *27.68* | 49.88 | 53.50 | 38.76 | 36.11 |
| RemoteCLIP | ✓ | | ViT-B/32 | 38.59 | 22.76 | 51.21 | 53.27 | 34.39 | 38.42 |
| SeCo | ✗ | Image | ResNet-50 | 45.63 | 23.86 | 68.78 | 78.42 | 40.48 | 51.11 |
| CACo | ✗ | | ResNet-50 | 48.50 | 26.92 | *69.90* | *79.15* | *47.40* | **57.02** |
| Satlas | ✓ | | Swin | 42.04 | 18.47 | 63.64 | 72.61 | 29.17 | 30.46 |
| **GRAFT** | ✗ | Text | ViT-B/32 | 47.80 | 29.31 | 66.12 | 72.36 | 45.88 | 45.35 |
| **GRAFT** | ✗ | | ViT-B/16 | **63.76** | **32.46** | **81.56** | **85.21** | **49.61** | *53.86* |

## 4.1 IMAGE-LEVEL UNDERSTANDING

We are interested in understanding how GRAFT models performs at image-level understanding tasks. We consider two tasks: zero-shot image classification (i.e., assign an image a text label) and text-based image retrieval (i.e., retrieve all images relevant to a text query).

**Datasets and Evaluation.** For models trained on Sentinel-2 data, we evaluate classification and retrieval performance on EuroSAT (Helber et al., 2019) and BigEarthNet (or BEN by Sumbul et al. (2019)). For models trained on NAIP data, we evaluate zero-shot classification on SAT-4 and SAT-6 (Basu et al., 2015). We also create a multi-label NAIP dataset using annotations from OpenStreetMap (OSM contributors, 2023) to evaluate both classification and retrieval.

**Baselines.** We consider CLIP as a natural baseline. There is no prior work on vision-language models trained without annotations. Instead, we consider two vision-language models that are trained in a *supervised* manner with text annotations: CLIP-RSICD (Arutiunian et al., 2021) and RemoteCLIP (Liu et al., 2023). Please refer to Appendix B for the text prompt used for the zero-shot models.

In addition to the aforementioned zero-shot baselines, we also provide one-shot baselines on various representation learning approaches for Sentinel-2 models. For these baselines, instead of using textual queries, we use the feature of the single example. To get a better understanding of the performance, we report the average performance over 10 different runs.

**Results.** For the performance of GRAFT models trained on Sentinel-2 data, we report classification results (left) and retrieval results (right) in Tab. 1; for GRAFT models trained on NAIP, we report classification and retrieval performance in Tab. 2. Almost across the board (with the exception of one retrieval metric), our model (GRAFT) outperforms all other models irrespective of pre-training modality and pre-training supervision. Furthermore, the margin of improvement is large for many of the datasets. For example, for EuroSAT classification, GRAFT with VIT-B/16 outperforms all other models by almost **10 points (20% relative)**. The gains on SAT-6 are even larger (**30 points**). In general, the VIT-B/16 backbones perform better, with the exception of SAT-4/SAT-6. The latter two datasets have $28 \times 28$ images (smaller than the VIT-B/32 patch), which may explain this discrepancy.

The significant improvement compared to *supervised* VLMs such as CLIP-RSICD and RemoteCLIP suggests that it is much more advantageous to use many ground-satellite image pairs without text, rather than fine-tuning in a supervised manner on small datasets.

In addition, our *zero-shot* GRAFT models also outperform *one-shot* performance of previous foundation models, including Satlas which is pre-trained on 302 million *labeled* examples. For example, on the challenging BigEarthNet, zero-shot GRAFT VIT-B/32 is able to outperform 1-shot Satlas by $\sim 13$ mAP on average without using any annotated images.

Table 2: Zero-shot performance of NAIP image-level model on classification (Left) and retrieval (Right) metrics. GRAFT outperforms the existing models on both retrieval and classification tasks at this resolution as well (red and blue indicate best and second best performance).

| Model | Backbone | Classification | | | Retrieval | |
| | | SAT-4 Acc. | SAT-6 Acc | NAIP-OSM mAP | NAIP-OSM mAP$^{100}$ | mAP$^{20}$ |
|---|---|---|---|---|---|---|
| CLIP | ViT-B/32 | *50.74* | 30.39 | 26.94 | 56.71 | 57.51 |
| CLIP | ViT-B/16 | 46.60 | 25.82 | 31.89 | 60.88 | 61.59 |
| CLIP-RSICD | ViT-B/32 | 47.72 | *40.57* | *32.51* | *64.58* | *63.71* |
| RemoteCLIP | ViT-B/32 | 33.55 | 20.76 | 23.29 | 42.03 | 40.84 |
| **GRAFT** | ViT-B/32 | **66.64** | **72.42** | 41.36 | 74.31 | 74.77 |
| **GRAFT** | ViT-B/16 | 53.42 | 66.57 | **42.47** | **75.35** | **76.55** |

Table 3: Accuracy of our pixel-level models on the Satlas land-cover segmentation task.

| Model | NAIP | Sentinel |
|---|---|---|
| CLIPSeg | *26.99* | *20.02* |
| **GRAFT** | **49.38** | 31.95 |
| **GRAFT+SAM** | - | **32.43** |

Table 4: Performance of our model on zero-shot VQA when used with ViperGPT.

| Model | Question type | | | | Average |
| | Presence | Area | Comp. | Count | |
|---|---|---|---|---|---|
| GLIP | **72.04** | *22.91* | *48.98* | *23.57* | *37.69* |
| GRAFT | *61.46* | **32.04** | **50.34** | **44.38** | **44.05** |

## 4.2 PIXEL-LEVEL UNDERSTANDING

We next evaluate our pixel-level model on zero-shot segmentation on the Satlas (Bastani et al., 2023) segmentation task. This task offers two benchmarks: one at NAIP resolution and the other at Sentinel-2 resolution. The segmentation task is 11-way landcover classification, but we remove 3 of the 11 categories that do not appear at all in the test set.

For the NAIP resolution, the segmentation masks are at a resolution 16 times smaller than the images. As such, for the NAIP resolution benchmark, we directly evaluate GRAFT without any SAM-based refinement. For the Sentinel-2 resolution, we report results both with and without SAM.

**Baseline:** We compare our model against CLIPSeg (Lüddecke & Ecker, 2022), which train a decoder on top of CLIP (using additional internet images) to perform open-world segmentation.

**Results:** Tab. 3 shows the results. GRAFT nearly **doubles** the per-class accuracy on the NAIP benchmark, and yields more than a **50%** relative improvement on the Sentinel-2 benchmark. Interestingly, refinement with SAM offers only a minor improvement.

## 4.3 VISUAL QUESTION ANSWERING

ViperGPT uses GLIP (Li et al., 2022) as the underlying vision model. We swap out the GLIP model with our own pixel-level model trained on the NAIP data and evaluate both models on the test set of the high-resolution version of RSVQA benchmark (Lobry et al., 2020). We subsample 500 unique test questions (because of API restrictions on the ViperGPT's LLM) and replace the images with NAIP images from the same location. We report our results in Tab. 4. With our pixel-level model, ViperGPT outperforms the variant with GLIP by a significant margin across all question and question types (except presence), yielding an almost **6 point** improvement in zero-shot accuracy. Fig. 2 (last row) shows example queries, where GLIP fails to detect the swimming pool or bike path and therefore gets an incorrect answer. However, the GRAFT model can get the correct answer.

## 4.4 ABLATIONS

We conduct the following two ablations to validate our design choices. Reported performance is on a separate held-out validation set of NAIP images annotated with Open Street Maps (see Appendix C.2 materials for more information about this validation set).

**Importance of sampling in data collection:** To reduce overlapped satellite images, we sample satellite images such that the central pixels of each satellite image are at least 112 meters apart (see

Sec. 3.2). If instead we sample the satellite images randomly, performance drops from 74.8% mAP to 69.3% mAP. This vindicates our sampling strategy.

**Importance of images vs text:** Instead of using the CLIP embedding of internet images as the intermediary for learning, we could instead use alt-text information from the internet images as direct textual supervision for contrastive learning. Replacing the ground image embeddings with the embeddings of this alt-text reduces performance from 74.8% mAP to 65.4% maP. Even using textual embeddings as an auxiliary loss still yields reduced performance (70% mAP). This suggests that images, not text, are the better intermediary for alignment, possibly due to the large noise in alt-text (see Appendix E for more details). Further ablations can be found in the Appendix E.

## 4.5 EXAMPLE APPLICATIONS

Our models can be used to create powerful visualization and analysis tools for city and regional planning, agriculture, and climate science. For example, we can use our model to query for any open-world concept on satellite images from a large region and create maps for that concept. We show example maps for cities, roads and farms in Massachusetts using NAIP images in Fig. 5.

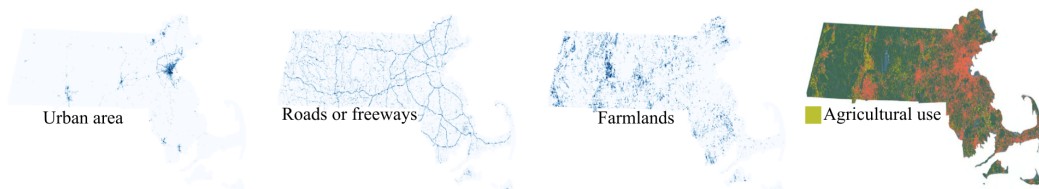

Figure 5: Density maps produced using open-world queries for cities, roads, and farmlands using our method (darker blue means higher density). The right-most map shows the true agricultural land use pattern. Our map matches with the ground truth.

## 5 DISCUSSION AND CONCLUSION

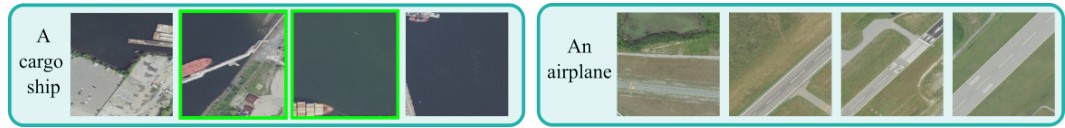

Figure 6: Top retrievals of GRAFT when finding dynamic objects such as cargo ships or airplanes. Images with green box show a successful retrieval of dynamic objects.

**Limitations:** One limitation of GRAFT is that it is difficult to capture dynamic objects in satellite images. Our models cannot align satellite images with internet images of dynamic objects because of the low revisit rate of satellites. Fig. 6 shows top retrieval for concepts like 'cargo ship'. While GRAFT cannot find such dynamic objects, it can still retrieve where they might potentially be found. Using a higher temporal resolution (daily revisit) is a way to handle this problem in the future. Please refer to the Appendix F for more potential directions of improvement.

**Conclusions:** We present GRAFT— a method to train vision-language models for satellite images that can be used for a diverse set of vision problems in remote sensing. To train this model, we propose a novel method of connecting satellite images with text using internet images as an intermediary. This allows us to train a first-of-its-kind large-scale VLM for remote sensing imagery without expensive language-paired data. This VLM yields state-of-the-art accuracy for zero-shot image classification, retrieval, segmentation and VQA, providing gains of up to 20% on classification and up to 80% for segmentation. We believe that this VLM can enable scientists of all stripes to analyze the vast trove of satellite data that is now available.

## 6 ETHICS

As in all visual recognition, there is potential negative impact through violations of individual privacy. While the resolution of our satellite imagery is not sufficient for identifying individuals, it does hold the potential for detecting environmental alterations made by individuals. The use of our proposed techniques for surveillance should be appropriately regulated. Furthermore, models such as CLIP learn various biases relating to race, culture, gender, and more from the internet (Agarwal et al., 2021). Since GRAFT is trained with the text image encoder of CLIP, it may inadvertently perpetuate or even exacerbate such biases. The use of our model should be done carefully in applications that are sensitive to such biases.

Furthermore, to mitigate the misuse of our models and data we release the data and the trained models only to people complying with an ethics agreement. By adhering to the ethics agreement, the user agrees to follow the laws, regulations, and the ASPRS Code of Ethics (ASPRS). We also ask the user to provide their intended use case before accessing the model. Another thing to note is that the data we use for this training is already freely and publicly available, hence our model is not exacerbating the issue of privacy.

## 7 ACKNOWLEDGMENTS

This research is based upon work supported in part by the Office of the Director of National Intelligence (Intelligence Advanced Research Projects Activity) via 2021-20111000006, the NSF STC for Learning the Earth with Artificial Intelligence and Physics, and the U.S. DARPA ECOLE Program No. #HR00112390060. The views and conclusions contained herein are those of the authors and should not be interpreted as necessarily representing the official policies, either expressed or implied, of ODNI, IARPA, DARPA, or the US Government. The US Government is authorized to reproduce and distribute reprints for governmental purposes notwithstanding any copyright annotation therein.

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

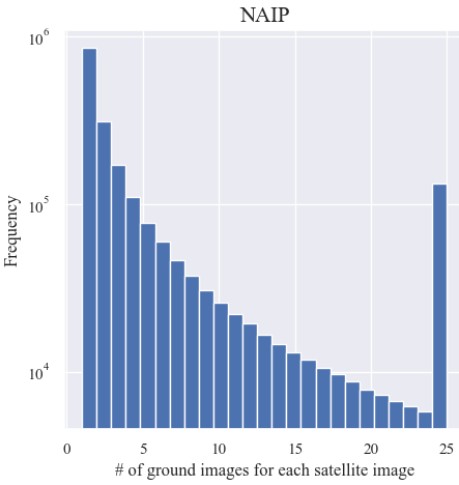
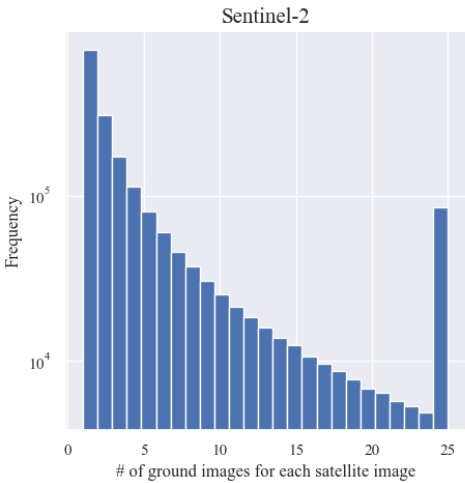

Figure 7: Distribution of the number of images corresponding to a single satellite images in our NAIP (Left) and Sentinel-2 (Right) datasets.

## A    TRAINING DATASET DETAILS

The dataset at NAIP resolution contains 10.2 million pairs of satellite-ground image pairs. It contains 2.0 million unique satellite images and 7.9 million unique ground images. Note that while the overlap between satellite images is minimal they are not completely non-overlapping. As a result, the same ground image can be a part of two different satellite images in our training data. Fig. 7 (left) shows the distribution of ground images for each NAIP image in our dataset. The distribution of the number of ground images per satellite image is a decreasing function. The number of satellite images with 25 images is much higher than the rest because we subsample 25 images whenever a satellite image contains more than 25 ground images.

**NAIP Preprocessing:**    We download the RGB bands of NAIP images. Since NAIP imagery is already stored in a normalized fashion we do not need to take anymore additional steps.

The dataset at Sentinel-2 resolution contains 8.7 million pairs of satellite-ground image pairs. It contains 1.9 million unique satellite images and 7.6 million unique ground images. Fig. 7 (right) shows the distribution of ground images for each Sentinel-2 image which follows the same trend as NAIP dataset. Fig. 8 shows example of paired data from our NAIP and Sentinel-2 dataset.

**Sentinel-2 Preprocessing:**    We download the B4, B3, B2 (corresponding to RGB) bands of Sentinel-2 images. Note that the data from Sentinel-2 is not coming from an RGB camera sensor. The values in B4, B3, and B2 capture the reflectance for a specific wavelength close to Red, Green, and Blue. We downscale these intensities by 3000 in order to get images that approximate RGB camera sensors (see Fig. 8 (bottom)). This is a standard practice followed by other works as well(Mall et al., 2023; Manas et al., 2021).

### A.1    SPATIAL AND TEMPORAL ALIGNMENT

**Temporal alignment:**    The temporal revisit for Sentinel-2 is about a week. For every Flickr image, we collect the temporally closest cloudless satellite image (see Sec. 3.2). The Flickr images in our dataset go back to 2014 while the earliest sentinel images we can get are from 2017. For the first three years of Flickr data, we sample the closest seasonally aligned images. On average the temporal gap between a Flickr image and a Sentinel image is around 21 days. So while it is difficult to connect ground images with satellite images of the same day, we can still align seasonality.

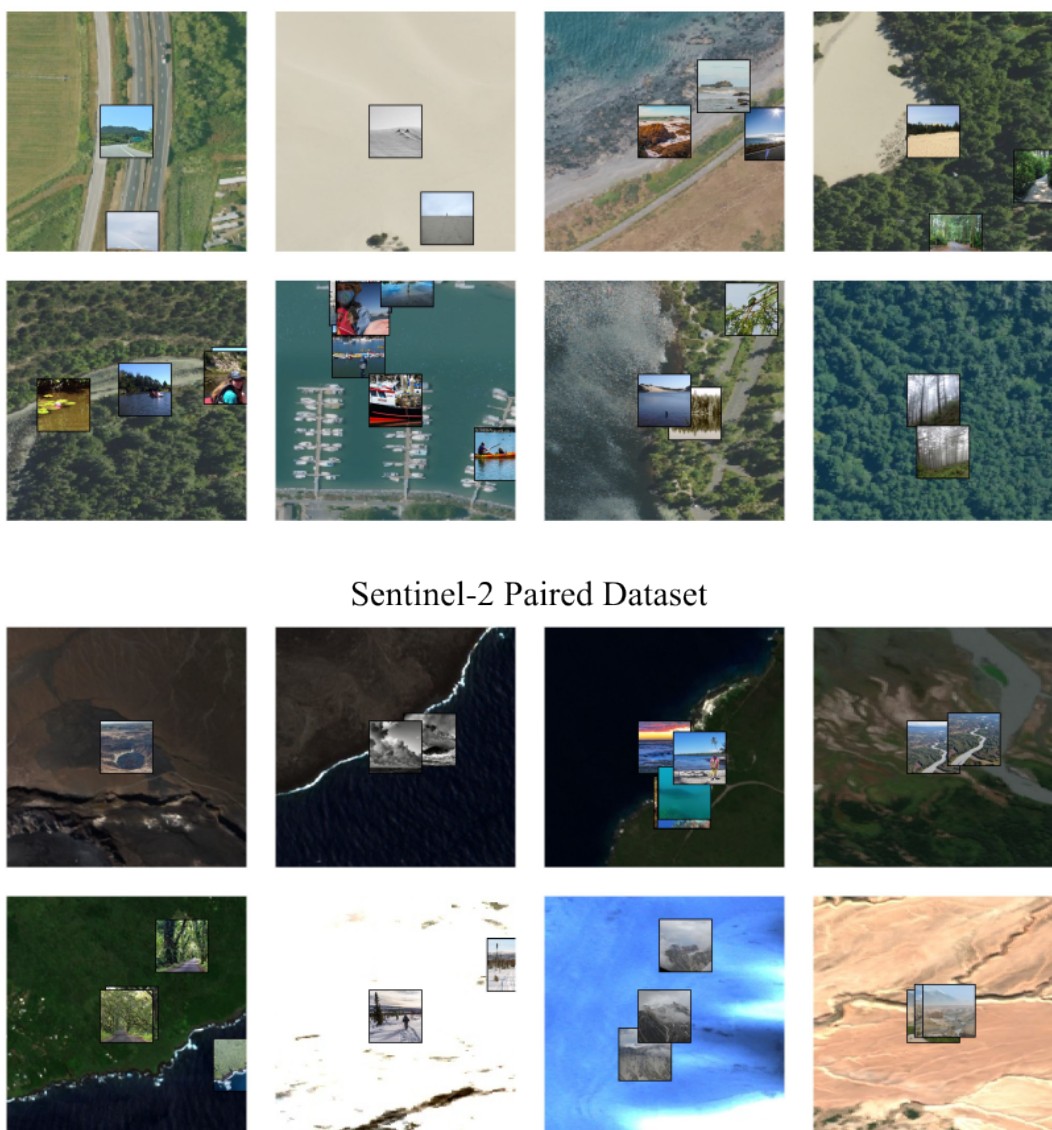

Figure 8: Examples of pairs from our NAIP and Sentinel-2 dataset. Each satellite image can contains multiple ground images. The ground images are shown at their corresponding location within the satellite images. NAIP (top) and Sentinel-2 (bottom).

**Spatial alignment:** As stated in Sec. 3.2, the Flickr API also provides accuracy in the geotags along with the geotags. We only use the images with the highest level of geotag accuracy (street level). However, it is not unreasonable to think that some of these geotags can still be incorrect. There are two reasons why we believe our model is robust to such noise. First, prior work on image-text contrastive learning such as OpenCLIP (Cherti et al., 2023) has shown that the training loss and architectures for such vision language models are robust even at a lower signal-to-noise ratio. Second, the patches we contrast image features with are big enough (14m for NAIP and 140m for Sentinel-2) that our model is invariant to a small amount of noise in geotagging.

## B  PROMPTING

Like CLIP, we test our models using template prompts such as `a photo of a {label}`. For classification metrics, with labels $l_j \in \mathbf{L}$, an image $x$ can be classified as label $l_{j*}$,

$$j^* = \arg\max_j f_S^I(x).f_T(\texttt{a photo of a } \{l_j\}) \tag{4}$$

In practice, we use multiple prompts and use the average text representation from those. For GRAFT we use the following prompts: `A photo of a {label}`, `A photo taken from inside a {label}`, `I took a photo from a {label}`. For baselines, we use the prompts prescribed in CLIP: `A centered satellite photo of {label}`, `A centered satellite photo of a {label}`, `A centered satellite photo of the {label}`.

Note that since our model is trained with internet images as an intermediary, it performs better with prompts describing internet imagery such as: `a photo of a {label}`. Whereas CLIP performs better with prompts describing satellite images such as `a satellite image of a {label}`.

## C  ADDITIONAL IMPLEMENTATION DETAILS

### C.1  TRAINING DETAILS

Regardless of the training datasets (NAIP or Sentinel-2), we train all models for 10 epochs using AdamW with weight decay set to 1e-2. For image-level model, we linearly ramp up the learning rate from 0 to 1e-5 and then decrease the learning rate using a cosine schedule. For pixel-level model, we linearly ramp up the learning rate from 0 to to 5e-5 and then decrease the learning rate to zero using a cosine schedule. All models are initialized using CLIP's weights and the temperature hyperparameter is set to $\tau = 0.07$.

### C.2  INTERNAL VALIDATION DATASET

For the development of the image-level models, we collected an internal validation set using Open-StreetMap (OSM contributors, 2023). This validation set contains 14 categories with a total of 2632 single-label images. We measured the performance of our image-level models on this dataset using mean average precision. For developing pixel-level model, we use a subset of NAIP-OSM we collected (see Appendix C.3). This subset contains 32 categories with 17k images. We select the best performing pixel-level models based on the zero-shot accuracy of patch prediction.

### C.3  IMAGE-LEVEL UNDERSTANDING EVALUATION details.

We measured our image-level model on 5 different datasets. The low-resolution Sentinel-2 model is evaluated on EuroSat and BigEarthNet Datasets.

**Eurosat:** We use the validation set of EuroSAT as our test set (the real test set is not public). It contains 5400 64×64 images from Sentinel-2 imagery from Europe. Each image is labeled with a single label out of 10 classes.

Table 5: List of 33 categories in the proposed NAIP-OSM dataset.

| | | | | |
|---|---|---|---|---|
| airport | football field | baseball field | beach | bridge |
| cemetery | commercial area | dam | equestrian facility | farmland |
| forest | garden | golf course | highway | marina |
| parking garage | park | parking lot | pond/lake | railroad |
| residential area | river | roundabout | sand area | school building |
| shooting range | soccer field | supermarket | swimming pool | tennis court |
| university building | warehouse | wetland | | |

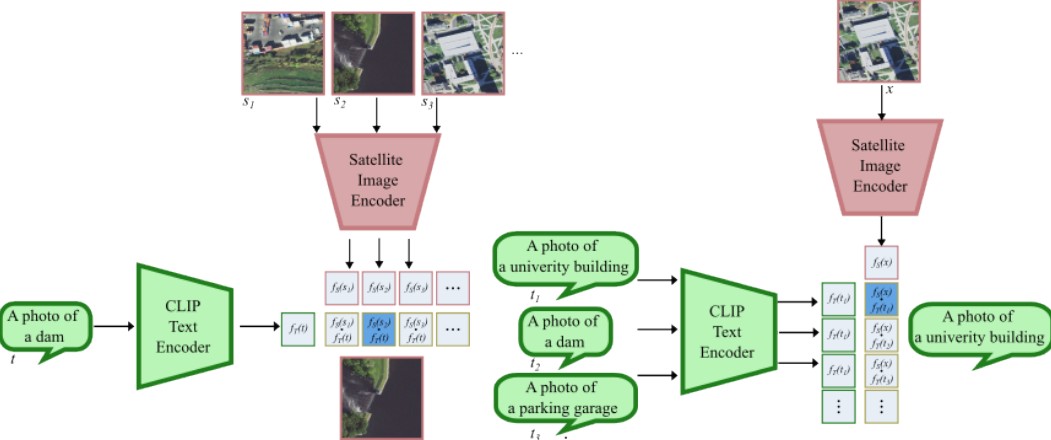

Figure 9: Inferring for text-to-image retrieval (left) and zero-shot classification (right) tasks with GRAFT.

**BigEarthNet (BEN):** BigEarthNet (Sumbul et al., 2019) is a 19-class multilabel classification dataset. The test set contains 104k 120×120 image from Sentinel-2.

The high-resolution NAIP model is evaluated on 3 datasets.

**Sat-4 and Sat-6:** The test set of the SAT-4 and SAT-6 datasets contains 64k and 81k 28×28 images respectively. SAT-4 and SAT-6 are labeled with 4 and 6 categories respectively on NAIP images. The fourth category of SAT-4 dataset is 'None', which is not suitable for zero-shot evaluation, so we evaluate with the remaining three categories only.

**NAIP-OSM:** Finally, since there are not a lot of evaluation benchmarks on NAIP images. We leverage OpenStreetMap data to create a large-scale multi-label classification and retrieval dataset for NAIP. This dataset contains 33 category labels and 1.7 million 224×224 NAIP images. Tab. 5 shows the list of categories in this dataset.

Since for many of these benchmarks the images are smaller than the resolution our models and baselines support. We try a variety of image pre-processing and report the best number for each method. We try zero padding, reflection padding, and resizing transforms and report the best numbers for each method. CLIP and CLIP-RSICD work best with resizing to 224×224 images. For GRAFT ViT-B/16 we use zero-padding as it results in the best performance. For RemoteCLIP, CACo, and SeCo, we use the pre-processing proposed by them. For Satlas, reflection padding works the best. When evaluating GRAFT ViT-B/32 on EuroSat, we had to pad in images in a way that the 64×64 images align with the input patches of ViT-B/32.

Fig. 9 illustrates how we use our model for text-to-image retrieval and zero-shot classification on these 5 benchmarks.

For multiclass classification (EuroSAT), we report the top1 accuracy. For the multi-label classification task (BigEarthNet), we report mean average precision (area under precision-recall curve). For retrieval tasks, we report the standard metric used for evaluating ranking: mAP@100 and mAP@20.

## C.4 Segmentation

Our segmentation results are evaluated on the newly proposed SatlasPretrain (Bastani et al., 2023) segmentation task. SatlasPretrain contains 2 test sets for NAIP and Sentinel-2 images. For NAIP, we use the test set along with the 2020 NAIP imagery. This results in a total of about 39k 512×512 NAIP images. The segmentation labels consist of 11 landcover classes (3 of which never occur in the test set). We evaluate the per-class accuracy on the remaining 8 classes. While the NAIP images are of size 512×512, the landcover labels are at a lower resolution of 32×32. With a ViT-B/16 model, this allows us to evaluate patch prediction without having to upscale either using interpolation or other bottom-up segmentation models such as SAM. For evaluation, we divide the 512×512 images into 4 224×224 and pass them through our models and baselines.

For Sentinel-2, we use the test set along with the *sentinel-2-small* imagery. This results in a total of about 2183 512×512 Sentinel-2 images. Same as before we evaluate with 8 out of 11 classes. Similiar to NAIP, for evaluation, we divide the 512×512 images into 4 224×224 and pass them through our models and baselines.

However, for Sentinel-2, the landcover masks are also of size 512×512. Therefore we evaluate GRAFT in two ways. Firstly, we evaluate the performance by upscaling the logits (bicubic interpolation) and using the maximum logit value as a prediction. This gives us the performance of the standalone GRAFT model. Alternatively, we also combine our model with SAM. In order to perform semantic segmentation, we use SAM's segment everything mode to get bottom-up segments. Then we assign each segment the class corresponding to the majority of the pixels in it. Our experiments in Sec. 4 suggest that using SAM in leads to a minor improvement in performance, suggesting the standalone GRAFT can also perform good segmentation.

## C.5 VQA

As stated in the main paper, we conducted VQA evaluation with 500 unique questions from the high-resolution RSVQA testset. Note that where there are 500 unique questions, the number of image-question-answer tuples is much larger since the same question can be asked on multiple images. In total, there are 2985 image-question-answer tuples in the evaluation set (presence: 576, Area: 1161, Comparison: 590, Count: 658).

## D Additional Results

**Open Vocabulary Image Retrieval**   Fig. 10 shows more examples of open-world concepts and the understanding abilities of our model. Our method can understand and retrieve several interesting open-world concepts. For example, it can retrieve very niche concepts such as campgrounds or hedge mazes. GRAFT can also be used for tracking concepts related to sustainable development. For example, we can detect solar farms, water treatment facilities, and wind farms (Fig. 2). Such use cases would be very useful to scientists interested in analyzing a region and tracking, for example, renewable sources of energy.

**Multi-spectral Satellite Images**   The main contribution of this work is to demonstrate that we can train large-scale vision language models for satellite images without direct text annotations, by using ground images as an intermediary. Therefore, we can also use this technique to train a model for multi-spectral satellite images. To show this, we train another VIT-B/16 model on 100k multi-spectral sentinel images-ground image pairs. To account for the change in input channels, we change the patch embedding input dimension from 3 to 12. The weights for the RGB channels are initialized to CLIP's weights, and the rest are set to zero. On the EuroSat multi-spectral test set, this model leads to a zero-shot classification accuracy of 53.78, which is an improvement of 1.26% on the RGB model (52.52%) trained with the same set of images. This shows that our insight generalizes to multi-spectral datasets.

## E Ablations

We provide additional ablations on GRAFT in this section.

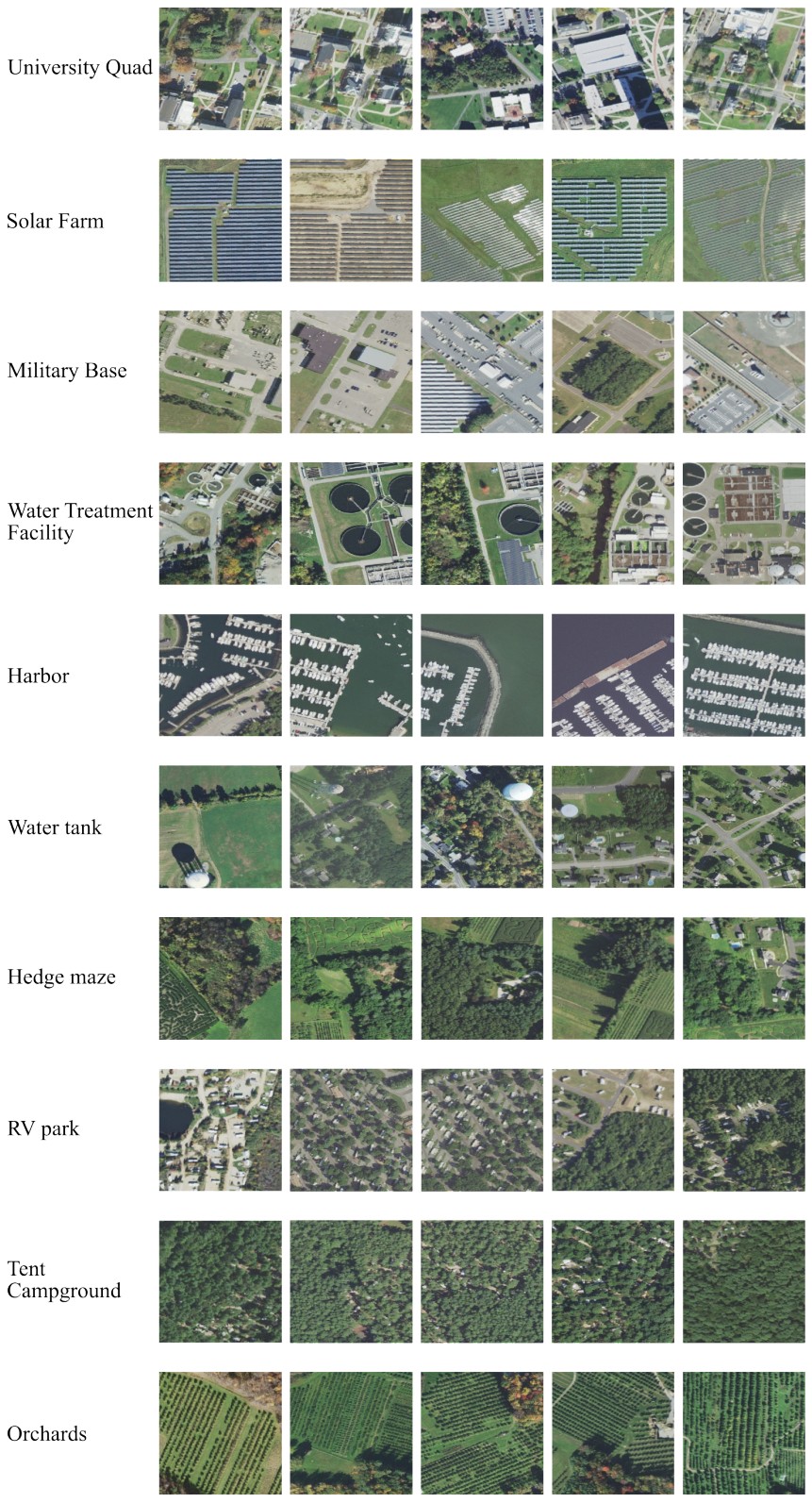

University Quad

Solar Farm

Military Base

Water Treatment Facility

Harbor

Water tank

Hedge maze

RV park

Tent Campground

Orchards

Figure 10: More examples of open-world text-to-image retrieval with GRAFT. Our model can understand various fine-grained concepts such as a water treatment facility or a hedge maze and less concrete concepts such as a university quad.

### E.1 ABLATIONS: ALT-TEXT ALIGNMENT

In Sec. 4.4 we show that ground-satellite image alignment results in better-performing models than aligning text to satellite images. We believe the main factor for this is that the ground images can provide much better semantic features to the satellite image encoder for alignment.

We use the captions and tags from Flickr data as the alt-text annotations. Often these captions include less meaningful information such as the filenames ("1819-img.png") automatically uploaded from phones/cameras. Other times the captions are very specific and do not describe the scene in the image, e.g. "A photo from the island where grandma grew up". In both these cases, the captions do not correctly capture the right level of semantic information and therefore the images are more helpful.

The image-text pairs in other datasets such as LAION-5B Schuhmann et al. (2022) also contain such data. However, the methods trained on it can still learn something useful due to the sheer scale of the dataset (5 billion vs 10 million). We posit that, to learn directly from text we might need a dataset with a satellite image-text pair at this scale, which is harder to obtain and train with.

### E.2 ABLATIONS: CLIP INITIALIZATION

We initialize our model from CLIP because this helps in better learning and faster convergence. We also tried training a satellite encoder from scratch however this model performed significantly worse than CLIP and Our Best model. The EuroSat zero-shot classification performance for a model trained from scratch is 42.46% vs CLIP (53.59%) and GRAFT (63.76%). Since all the baselines we compare against are also initialized from CLIP and are not trained from scratch this comparison is fair.

### E.3 ABLATIONS: LOSS FORMULATIONS

In this section, we explore different loss formulations that could be used to create a satellite image representation that aligns with CLIP's representation of the ground images. By default, GRAFT uses Eq. (3) to enforce the intuition that the network should produce a representation of the satellite image that is close to its ground images. A few other losses could achieve similar properties:

$$\mathcal{L}_2^I(\mathcal{B}, f_S^I) = \frac{1}{N_B} \sum_{i=1}^{N_B} -\log \frac{1}{N_i} \sum_{j=1}^{N_i} \frac{\exp(f_S^I(s_i) \cdot f_G(g_i^j)/\tau)}{\sum_{a=1}^{N_B} \sum_{b=1}^{N_i} \exp(f_S^I(s_i) \cdot f_G(g_a^b)/\tau)} \tag{5}$$

$$\mathcal{L}_3^I(\mathcal{B}, f_S^I) = \frac{1}{N_B} \sum_{i=1}^{N_B} -\log \frac{\exp((f_S^I(s_i) \cdot (\bar{z}_i/||\bar{z}_i||))/\tau)}{\sum_{a=1}^{N_B} \exp(f_S^I(s_i) \cdot (\bar{z}_a/||\bar{z}_a||)/\tau)} , \bar{z}_i = \frac{1}{N_i} \sum_{j=1}^{N_i} f_G(g_i^j) \tag{6}$$

$$\mathcal{L}_4^I(\mathcal{B}, f_S^I) = \frac{1}{N_B} \sum_{i=1}^{N_B} \frac{1}{N_i} \sum_{j=1}^{N_i} ||f_S^I(s_i) - f_G(g_i^j)||_2^2 \tag{7}$$

We experiment with all these losses on the NAIP high-resolution dataset and report its performance on the internal validation set (Appendix C.2) in Tab. 6. Even though Khosla et al. (2020) have shown that $\mathcal{L}_2^I$ is suboptimal for their supervised setup, we find the performance to be on par with using the default formulation.

However, not all loss formulations are equal. Using the conventional contrastive loss with the average representations of all ground images as positive (Eq. (5)) yields a model that underperforms the GRAFT formulation. Discerning readers might also question whether it is necessary to enforce a satellite image's embedding to stay far from ground images associated with other satellite images. Performance using the l2 loss Eq. (7) suggests that not only it is important that a satellite image's embedding to stay close to all its associated ground images, it is also crucial that its embedding stays away from ground images associated to other satellite images.

### E.4 ADDITIONAL ABLATIONS: SCALING

GRAFT uses a large amount of satellite-ground image pairs to sidestep the need for textual annotations for training. To understand the behavior of GRAFT, we trained image-level VLMs using

Table 6: Performance of different image-level VLMs trained using different loss formulations to align the satellite image modality with the ground image modality. For the precise definition of the loss functions, please refer to Appendix E.3

| Formulation | mAP |
|---|---|
| $\mathcal{L}_2^I$ | 74.76 |
| $\mathcal{L}_3^I$ | 71.42 |
| $\mathcal{L}_4^I$ | 66.80 |
| Default $\mathcal{L}^I$ | 74.78 |

various amounts of satellite images and reported the performance in Fig. 11. We observe two different behaviors with images with different resolutions. For the model trained on NAIP images (left), the performance platues after $2 \times 10^5$ examples whereas the model trained on low-resolution sentinel-2 images (right) continues to scale with more data. We conjecture that the performance of NAIP models plateus because NAIP only covers the United States and a small amount of satellite images is enough to learn a good embedding.

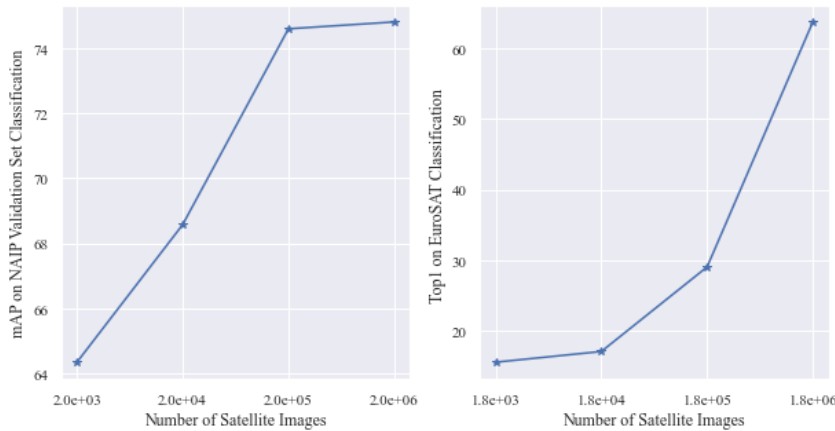

Figure 11: Performance of the Image-level VLMs trained on various amounts of satellite images using GRAFT. Left: a ViT-B/16 VLM trained on high-resolution NAIP images and evaluated on our internal validation set. Right: a ViT-B/16 VLM trained on low-resolution sentinel-2 images and evaluated on EuroSAT classification. The performance for the high-resolution model plateaus after $2 \times 10^5$ examples whereas the low-resolution Sentinel-2 model continues to scale with more data.

## F  FUTURE WORK

A limitation of GRAFT and other CLIP-like VLMs is that they do not possess text-generation capabilities such as BLIP-2 (Li et al., 2023). Exploring how to use ground images as an intermediary for text generation is an interesting avenue to explore in the future. However, as our results show, we can combine our model, with other frameworks– for example with ViperGPT– to solve tasks such as VQA. In the future, we can also combine GRAFT with other frameworks such as ClipCap (Mokady et al., 2021) to get more powerful VLMs with more capabilities such as captioning.

Another limitation that might arise since we do not fine-tune the text encoder, is that the text encoder might be biased towards concepts of ground images. For example, directional concepts such as "left of" or "top of" might not correspond meaningfully in the satellite images. Despite this aspect, our method performs better than other vision-language models (some of which are supervised with text). This shows that while there might be biases, our model is robust to them for most remote sensing recognition tasks. Nonetheless, it would be interesting for future work to look into this issue and create stronger models. For instance, one possible solution for this could be to fine-tune with a small amount of satellite image-caption data.

