# OpenReview forum: "Remote Sensing Vision-Language Foundation Models without Annotations via Ground Remote Alignment"
_ICLR.cc/2024/Conference — ICLR 2024 poster_

### Official Review · Reviewer_J5AT · 2023-10-14

**Soundness:** 3 good
**Presentation:** 3 good
**Contribution:** 2 fair
**Rating:** 6
**Confidence:** 2

**Summary:**

The authors used internet images to train image encoders. The proposed strategy can efficiently train visual-language models and does not introduce additional annotations. The authors verified the effectiveness of the proposed strategy on multiple subtasks such as zero-shot, open-vocabulary image classification, retrieval, segmentation and visual question
answering for satellite images.

**Strengths:**

The motivation is sound, and the proposed method is theoretically feasible. Experimental results show that the framework achieves good performance.

**Weaknesses:**

1. The proposed method is not innovative enough. I'm not an expert in this field, so I'm not sure about it.
2. The images in the manuscript are of such low resolution that they are difficult to read.

**Questions:**

N/A

---

> ### Author Response · Authors · 2023-11-19
> **Response to reviewer J5AT**
>
> We thank the reviewer for their comments and address their specific concerns. All the new updates in the paper are highlighted in red.
>
> ---
>
> **Innovation (W1)**: Our work is the first scalable approach for learning a vision-language model for satellite images which *other reviewers* have appreciated. Moreover, to our knowledge, this is the first method using the idea of an intermediary representation to align two different modalities. We believe that our approach is novel and our results show that we can surpass all the existing baselines in performance.
>
> **Image size (W2)**: Thank you for this feedback! We have updated the figures with larger fonts and larger images wherever possible. We have also advised readers to refer to the digital version of the paper, wherever the font remains smaller. If some images are still small due to space constraints, please let us know and we can add larger images in the supplementary.

---

> ### Comment · Reviewer_J5AT · 2023-11-21
>
> All my concerns were resolved and this is a good paper.

---

### Official Review · Reviewer_U5BT · 2023-10-30

**Soundness:** 3 good
**Presentation:** 3 good
**Contribution:** 3 good
**Rating:** 6
**Confidence:** 4

**Summary:**

This paper develops remote sensing vision-language foundation models, named GRAFT, by aligning the embedding of satellite images to internet images in the CLIP space. The obtained models perform well in zero-shot classification, retrieval, segmentation and VQA tasks.

**Strengths:**

1. The authors notice the 1-to-N relationship between satellite image and ground image, and develop image-level and pixel-level models, respectively, by designing two kinds of contrastive losses.
2. The authors collected nearly ten million images for pretraining the model.

**Weaknesses:**

1. In Section 1, the authors claim existing remote sensing image-text datasets have only 10k samples. However, as far as I know, the volume of RS5M has reached 5 million.
2. According to Section 3.4, the model has been initialized by the CLIP image encoder. So, is it that the high performances in downstream tasks are from CLIP pretraining? It is suggested to train the satellite image encoder $f_S$ from scratch to show the real ability of the proposed method.
3. The authors claim that they developed a VLM without textual annotations. Alignment is a good idea (and probably a cheap solution) to obtain a remote sensing VLM by leveraging existing ones from other domains. However, experiments show that the proposed model is still served as a visual encoder in existing VLM models. GRAFT cannot independently finish vision-language tasks, especially for text generation (VQA, captioning, etc.). Indeed, since it is only aligned with the CLIP visual encoder, its cross-modality ability is limited by the pre-trained clip text encoder. A discussion of its limitations being applied to other VLMs should be provided.
4. While the authors have indeed discussed the domain gap between natural images and remote sensing images, discrepancies in how objects are represented and described in these two image types persist. For instance, the size of similar objects (e.g., cars) in natural images may undergo changes due to projection transformation, whereas it remains consistent in remote sensing images. Additionally, the orientation and object placement differ between these two image categories (front view versus top view). Consequently, given that the text descriptions used to train the CLIP model are biased toward natural images, the challenge lies in bridging the gap between describing natural images and remote sensing images, which may limit the usage of the proposed method.

4. Some relevant works are expected to be reviewed, which can offer readers a comprehensive overview of the current advancements in this field:
SatViT: Pretraining Transformers for Earth Observation, IEEE GRSL, 2022.
An Empirical Study of Remote Sensing Pretraining, TGRS, 2023.
RS5M: A Large Scale Vision-Language Dataset for Remote Sensing Vision-Language Foundation Model, arXiv preprint arXiv:2306.11300 (2023).
Advancing Plain Vision Transformer Toward Remote Sensing Foundation Model, TGRS, 2023.
RSGPT: A Remote Sensing Vision Language Model and Benchmark, arXiv preprint arXiv:2307.15266 (2023).

**Questions:**

See the weaknesses

---

> ### Author Response · Authors · 2023-11-19
> **Response to reviewer U5BT**
>
> We are grateful to the reviewer for their constructive feedback. We address the specific concerns:
>
> ---
>
> **RS5M (W1)**:  Thanks for bringing this to our attention — we have added RS5M to the related work. For your information, the RS5M dataset contains a subset of images that automatically filter from publicly available text-image datasets. As a result, the images come from a diverse range of sources and are not uniform. For example, even in the examples shown in the RS5M paper, we can see many images taken from a plane window. Additionally, our approach allows more control of the data collection process. For instance, we could collect sentinel-2 satellite data from spectral bands other than RGB, allowing us to construct a VLM that could be used on different spectral bands. RS5M, on the other hand, consists of mostly RGB images and thus could only be used for constructing a VLM for RGB images. In addition, the results released by the RS5M paper show that our model performs better on the EuroSat zero-shot classification task (**63.76%** vs 61.48%).
>
> **CLIP Initialization (W2)**: We initialize from CLIP because this helps in better learning and faster convergence. We also tried training a satellite encoder from scratch however this model performed significantly worse than CLIP and Our Best model. The zero-shot classification performance for a model trained from scratch is 42.46% vs CLIP (53.59%) and GRAFT (63.76%).
>
> Secondly, since all the baselines we compare against are also initialized from CLIP and are not trained from scratch, these comparisons are fair. We also compare our model to the CLIP baseline for fair evaluation. We have added this experiment in the supplementary.
>
> **Limited Text Generation Capabilities (W3)**: Yes we agree with the reviewer that the text generation capabilities of our model are a limitation. We used the term VLM to describe models such as CLIP and ALIGN[1] in our paper. We haved added this as a limitation of our work and this is certainly an interesting avenue to explore in the future. Additionally, as our results show, we can combine our model with other frameworks– for example with ViperGPT– to solve other tasks such as VQA. We can also combine our VLM with other frameworks such as ClipCap [2] to get more powerful VLMs with more capabilities such as captioning.
>
> **Domain Gap (W4)**: We agree with the reviewer that some text concepts might be biased toward ground images since we do not change our text encoder. Despite this aspect, our method is better than other vision-language models (some of which are *supervised with text*). This shows that while there might be biases, our model is robust to them for most remote sensing recognition tasks. Nonetheless, it would be interesting for future work to look into this issue and create stronger models (e.g. finetuning on RS5M). We have added this discussion in the revision.
>
> **Relevant Work (W5)**: We have added these related works in the revision. RS5M and RSGPT are relevant to our work. SaTViT and RVSA although not directly related, are important architectures in the remote sensing community.
>
> All the new updates in the paper are highlighted in red. We hope that this addresses the reviewer's concerns.
>
> ---
> ### References:
> [1] *Jia, Chao, Yinfei Yang, Ye Xia, Yi-Ting Chen, Zarana Parekh, Hieu Pham, Quoc Le, Yun-Hsuan Sung, Zhen Li, and Tom Duerig. "Scaling up visual and vision-language representation learning with noisy text supervision." In International conference on machine learning, pp. 4904-4916. PMLR, 2021.*
>
> [2] *Ron Mokady, Amir Hertz, and Amit H Bermano. ClipCap: Clip prefix for image captioning. CoRR,
> 2021.*

---

### Official Review · Reviewer_hnot · 2023-11-01

**Soundness:** 3 good
**Presentation:** 3 good
**Contribution:** 4 excellent
**Rating:** 8
**Confidence:** 5

**Summary:**

This paper is addressing a significant problem, i.e., how to obtain a VLM for remote sensing images. The difficulty lies in that training a CLIP-like model needs a large amount of image-text pairs; however, for remote sensing images (e.g., satellite images), there are few such image-text pairs, even no. This problem blocks the open-vocabulary downstream task in remote sensing.
This paper gives a smart solution, which adopts ground image as the intermediate representation. Aligning satellite image representation with ground image representation replaces the directly aligning satellite image representation and text representation when given a good alignment between ground image representation and text representation.
This idea is very novel and experimental results are promising.

**Strengths:**

- The idea of aligning satellite image representation and ground image representation is novel and is promising to train a genuine remote sensing CLIP model as a foundation.
- Experiments are conducted on many downstream tasks and they show superior performances.

**Weaknesses:**

The keys to this idea are a well-aligned ground image-language model and the ground-satellite alignment.
The first condition seems CLIP can be responsible for it. The main concern is the ground-satellite alignment.
I am satisfied with everything except satellite-"alt-text" alignment.
I cannot understand why directly aligning satellite-text representations is much worse than the way using intermediate representation (ground). Does your "alt-text information" include more errors? thus, this textual supervision is biased causing worse results.
I am very interested in this part, however, authors has mentioned it in one stroke.

**Questions:**

N/A

---

> ### Author Response · Authors · 2023-11-19
> **Response to reviewer hnot**
>
> We appreciate the reviewer’s valuable feedback and we are pleased that the reviewer finds our work to be “novel” and  “promising”, and the problem we aim to solve “significant”. We address their concern here.
>
> ---
> **Alt-text alignment**: The main reason why alt-text alignment performs worse than ground-satellite alignment is that the ground images can provide much better semantic features to the satellite image encoder.
>
> We use the captions and tags from Flickr data as the alt-text annotations.  Often these captions include less meaningful information such as the filenames (*'1819-img.png'*) automatically uploaded from phones/cameras. Other times the captions are very specific and do not describe the scene in the image, e.g. *“A photo from the island where grandma grew up”*. In both these cases, the captions do not correctly capture the right level of semantic information and therefore the images are more helpful.
>
> The image-text pairs in other datasets such as LAION-5B [1] also contain such data.  However, the methods can still learn something useful due to the sheer scale of the dataset. We posit that, to learn directly from text we might need a dataset with a satellite image-text pair at this scale, which is harder to obtain and train with. We have added this information in the revision. The new updates in the paper are highlighted in red.
>
> ---
> ### Reference
> [1] *Christoph Schuhmann, Romain Beaumont, Richard Vencu, Cade Gordon, Ross Wightman, Mehdi
> Cherti, Theo Coombes, Aarush Katta, Clayton Mullis, Mitchell Wortsman, et al. Laion-5b: An
> open large-scale dataset for training next-generation image-text models. In NeurIPS, 2022.*

---

### Official Review · Reviewer_FsSQ · 2023-11-04

**Soundness:** 4 excellent
**Presentation:** 3 good
**Contribution:** 4 excellent
**Rating:** 8
**Confidence:** 4

**Summary:**

This paper presents a new method to train vision-language models for remote-sensing images without using any textual annotations. Authors use GGB bands of 2 types of satellite imagery: Sentinel-2 (hight resolution, 10m per pixel) and NIAP dataset (1 m per pixel). The model is trained on geotagged annotated photos, obtained from Flickr to train the models. The model itself consists of 2 parts: image-level and pixel level components. The result represent a large vison-language model with few example applications, such as annotation of the imagery and question-answering capability.

**Strengths:**

The paper represents an excellent example of the practically useful application, with a large curated dataset and e few useful applications, such as annotation and question-answering capabilities that will contribute well to the VLM literature.

**Weaknesses:**

Unfortunately neither the code not the dataset are available for review and the soundness of the paper cannot be well estimated.

One significant disadvantage of the paper is the fact that it is trained only on the RGB bands of the satellite imagery, significantly reducing applications in agriculture, earth and biological sciences.

**Questions:**

1) It's unclear whether there is a connection b/w ground and satellite imagery by timestamp (to monitor things in "almost real time" or to account for seasonality)
2) paragraph 3.2: you mention accurate geo-location. How do you measure/verify that the geolocation is accurate?
3) how do you account for typical errors in geotagging (~8m)
4) sec.3.4: do you check how old are the features on open street maps?
5) Ethics considerations: what measures can you take to avoid revealing personal details about the households/farms/properties of the individuals that can be observed with 1m satellite imagery?

**Details Of Ethics Concerns:**

1m resolution imagery might be potentially too revealing for private property, especially combined with LLMs and CLIP type models. Authors mention that their models should be used responsibly, but there are no measures made by paper authors themselves

---

> ### Author Response · Authors · 2023-11-18
> **Response to reviewer FsSQ**
>
> We thank the reviewer for their insightful comments and are glad that the reviewer likes the practical usefulness of our work. We address the specific comments here.
>
> ---
> ### Response to Weaknesses
>
> **Code and Dataset (W1)**: We promise to release the code and dataset upon acceptance.
>
> **Multispectral dataset (W2)**: The main contribution of this paper was to demonstrate that we can train large-scale vision language models for satellite images without direct text annotations, by using ground images as an intermediary. Nothing stops us from training a model with multispectral images. To show this, we train another VIT-B/16 model on 100k multispectral sentinel images-ground image pairs (To account for the change in input channels, we change the patch embedding input dimension from 3 to 12. The weights for the RGB channels are initialized to CLIP’s weights, and the rest are set to zero). On the EuroSat multispectral dataset, this model leads to a classification accuracy of 53.78, which is an improvement of **1.26%** on the RGB model trained with the same number of images. We have added this result to the supplementary.
>
> ---
> ### Response to Questions
>
> **Temporal alignment (Q1)**: The temporal revisit for sentinel-2 is about a week. For every Flickr image, we collect the temporally closest cloudless satellite image (see page 6). The Flickr images in our dataset go back to 2014 while the earliest sentinel images we can get are from 2017. For the first three years of Flickr data, we sample the closest seasonally aligned images. On average the temporal gap between a Flickr image and a sentinel image is around 21 days. So while it is difficult to connect ground images with satellite images of the same day, we can still align seasonality. We have provided this information in the revision.
>
> **Errors in spatial locations (Q2, 3)**: The Flickr API also provides accuracy in the geotags along with the geotags. We only use the images with the highest level of geotag accuracy. However, it is not unreasonable to think that some of these geotags can still be incorrect. Thankfully, prior work on image-text contrastive learning such as OpenCLIP has shown that the training loss and architectures for such vision language models are robust even at a lower signal-to-noise ratio. Secondly, the patches we contrast image features with are big enough (16m (in length) for NAIP and 160m for Sentinel-2) that our model is invariant to a small amount of noise in geotagging.
>
> **Open street maps (Q4)**: In section 3.4, we collect the latest NAIP images (2021 and 2022). We also collect the OSM data from the same year instead of the latest one. We have made this information clear in the revision.
>
> **Ethics Considerations (Q5)**: Thank you for raising these concerns! To mitigate the misuse of our model we plan to release the data and the trained models only to users complying with an ethics agreement. By adhering to the ethics agreement, the user agrees to follow the laws, regulations, and the ASPRS Code of Ethics. We also ask the user to provide their intended use case before accessing the model. Another thing to note is that the data we use for this training is already freely and publicly available, hence our model is not exacerbating the issue of privacy.
>
> ---
> The new updates in the paper are highlighted in red color.

---

### Meta-Review · Area_Chair_2baq · 2023-12-15

**Metareview:**

The paper presents a new method for training vision-language models for remote-sensing images without textual annotations. The authors use GGB bands of satellite imagery, Sentinel-2 and NIAP datasets, and geotagged annotated photos from Flickr. The model consists of image-level and pixel-level components, with applications in annotation and question-answering. The paper addresses the challenge of obtaining a VLM for remote sensing images, which require a large number of image-text pairs. The GRAFT model, developed by aligning satellite images to internet images in the CLIP space, performs well in zero-shot classification, retrieval, segmentation, and VQA tasks.

## Strengths

• Large curated dataset with useful applications like annotation and question-answering.
• Novel idea of aligning satellite and ground image representation for remote sensing CLIP model training.
• Superior performance demonstrated in experiments on multiple downstream tasks.
• Development of image-level and pixel-level models based on 1-to-N relationship between satellite and ground images.
• Pretraining model collected nearly ten million images.

## Weaknesses

• The paper is limited to training on the RGB bands of satellite imagery, limiting its applications in agriculture, earth, and biological sciences.
• The key to the model is a well-aligned ground image-language model and ground-satellite alignment.
• The main concern is ground-satellite alignment, which is criticized for its potential bias and worse results.
• The authors claim that existing remote sensing image-text datasets have only 10k samples, but the volume of RS5M has reached 5 million.
• The model was initialized by the CLIP image encoder, but it is suggested to train the satellite image encoder fS from scratch to demonstrate the method's real ability.
• The authors developed a VLM without textual annotations, but the proposed model is still used as a visual encoder in existing VLM models.
• The challenge lies in bridging the gap between describing natural images and remote sensing images, which may limit the usage of the proposed method.

**Justification For Why Not Higher Score:**

The paper proposes a nice approach but the audience is limited due to an specific problem.

**Justification For Why Not Lower Score:**

The paper is good enough to be rejected.

---

### Decision · Program_Chairs · 2024-01-16

Accept (poster)